

# Extinction and the U.S. Endangered Species Act

Noah Greenwald[1], Kieran F. Suckling[2], Brett Hartl[3] and
Loyal A. Mehrhoff[4]

[1] Center for Biological Diversity, Portland, OR, USA
[2] Center for Biological Diversity, Tucson, AZ, USA
[3] Center for Biological Diversity, Washington, DC, USA
[4] Center for Biological Diversity, Honolulu, HI, USA

## ABSTRACT

The U.S. Endangered Species Act is one of the strongest laws of any nation for preventing species extinction, but quantifying the Act's effectiveness has proven difficult. To provide one measure of effectiveness, we identified listed species that have gone extinct and used previously developed methods to update an estimate of the number of species extinctions prevented by the Act. To date, only four species have been confirmed extinct with another 22 possibly extinct following protection. Another 71 listed species are extinct or possibly extinct, but were last seen before protections were enacted, meaning the Act's protections never had the opportunity to save these species. In contrast, a total of 39 species have been fully recovered, including 23 in the last 10 years. We estimate the Endangered Species Act has prevented the extinction of roughly 291 species since passage in 1973, and has to date saved more than 99% of species under its protection.

## INTRODUCTION

Passed in 1973, the U.S. Endangered Species Act (ESA) includes strong protections for listed threatened and endangered species and has helped stabilize and recover hundreds of listed species, such as the bald eagle and gray whale (*Taylor, Suckling & Rachlinski, 2005*; *Schwartz, 2008*; *Suckling et al., 2016*). In part because of its strong protections, the ESA has engendered substantial opposition from industry lobby groups, who perceive the law as threatening their profits and have been effective in generating opposition to species protections among members of the U.S. Congress. One common refrain from opponents of the ESA in Congress and elsewhere is that the law is a failure because only 2% of listed species have been fully recovered and delisted (*Bishop, 2013*).

The number of delistings, however, is a poor measure of the success of the ESA because most species have not been protected for sufficient time such that they would be expected to have recovered. *Suckling et al. (2016)*, for example, found that on average listed birds had been protected just 36 years, but their federal recovery plans estimated an average of 63 years for recovery. Short of recovery, a number of studies have found the ESA is effectively stabilizing or improving the status of species, using both biennial status

Corresponding author
Noah Greenwald,
ngreenwald@biologicaldiversity.org

assessments produced by the U.S. Fish and Wildlife Service for Congress and abundance trends (*Male & Bean, 2005*; *Taylor, Suckling & Rachlinski, 2005*; *Gibbs & Currie, 2012*; *Suckling et al., 2016*).

In addition to recovering species, one of the primary purposes of the ESA is to prevent species extinction. Previous studies indicate the ESA has been successful in this regard (*McMillan & Wilcove, 1994*; *Scott et al., 2006*). As of 2008, the ESA was estimated to have prevented the extinction of at least 227 species and the number of species delisted due to recovery outnumbered the number of species delisted for extinction by 14–7 (*Scott et al., 2006*). In this study, we identified all ESA listed species that are extinct or possibly extinct to quantify the number of species for which ESA protections have failed and use these figures to update the estimated number of species extinctions prevented. This is the first study in over 20 years to compile data on extinction of ESA listed species, providing an important measure of one of the world's strongest conservation laws (*McMillan & Wilcove, 1994*).

## METHODS

To identify extinct or possibly extinct ESA listed species, we examined the status of all 1,747 (species, subspecies and distinct population segments) U.S. listed or formerly listed species, excluding species delisted based on a change in taxonomy or new information showing the original listing to have been erroneous. We determined species to be extinct or possibly extinct based on not being observed for at least 10 years, the occurrence of adequate surveys of their habitat, and presence of threats, such as destruction of habitat of the last known location or presence of invasive species known to eliminate the species.

To differentiate extinct and possibly extinct species we relied on determinations by the U.S. Fish and Wildlife Service, IUCN, species experts and other sources. In most cases, these determinations were qualitative rather quantitative. Species were considered extinct if surveys since the last observation were considered sufficient to conclude the species is highly likely to no longer exist, and possibly extinct if surveys were conducted after the last observation, but were not considered sufficient to conclude that extinction is highly likely (*Butchart, Stattersfield & Brooks, 2006*; *Scott et al., 2008*).

Source information included 5-year reviews, listing rules and critical habitat designations by the U.S. Fish and Wildlife Service (for aquatic and terrestrial species) or NOAA Fisheries (for marine species), published and gray literature, personal communication with species experts and classifications and accounts by NatureServe, IUCN and the Hawaiian Plant Extinction Prevention program. For each species, we identified year of listing, year last seen, NatureServe and IUCN ranking, taxonomic group, and U.S. Fish and Wildlife Service region. For species last seen after listing, we also searched for abundance estimates at time of listing in order to give a sense of likelihood of survival regardless of ESA protection.

Following previously developed methods, we estimated the number of species extinctions prevented by the ESA by assuming that listed threatened and endangered species have a comparable extinction risk to IUCN endangered species, which was estimated as an average of 67% over 100 years (*Mace, 1995*; *Schwartz, 1999*; *Scott et al., 2006*). We believe this estimate of extinction risk is conservative based on similarity of IUCN criteria to factors

considered in ESA listings, observed low numbers for species at time of ESA listing and observed correspondence between ESA listed species and species classified as endangered or critically endangered by the IUCN (*Wilcove, McMillan & Winston, 1993*; *Wilcove & Master, 2005*; *Harris et al., 2012*). Presumed extinction risk was then multiplied by the number of extant listed species and the proportion of a century in which species were protected by the ESA. Previous studies used the length of time the ESA has been in existence (1973-present) for the proportion of a century species have been protected (*Schwartz, 1999*; *Scott et al., 2006*), but because many species have not been protected the entire 45 years the law has existed, we instead used the more conservative average length species were protected (25 years). This corresponds to the following formula:

$$\text{Expected extinctions} = (spp. \times 100 \text{ year extinction risk} \times \text{average proportion of a century with protection}).$$

## RESULTS

We identified a total of 97 ESA listed species that are extinct (23) or possibly extinct (74). Of these, we found 71 extinct (19) or possibly extinct (52) species were last observed before they were listed under the ESA and thus are not relevant to determining the Act's success in preventing extinction (Table S1). These species were last seen an average of 24 years before protection was granted with a range of one to more than 80 years prior.

A total of 26 species were last seen after listing, of which four are confirmed extinct and 22 are possibly extinct (Table S2). On average, these species were last seen 13 years after listing with a range of 2–23 years. We were able to find an abundance estimate at the time of listing for 19 of these species, ranging from one individual to more than 2,000 with an average of 272. In several cases, these estimates were based on extrapolations from very few sightings.

The distribution of extinct and possibly extinct species was non-random with 64 of the 97 species from Hawaii and other Pacific Islands, followed by 18 from the southeast (Fig. 1). This was also the case for taxonomy. A total of 40 of the 97 species were mollusks dominated by Hawaiian tree snails and southeast mussels, followed by birds (18) and plants (17) (Fig. 2).

We identified several other species that have been missing for more than 10 years, but for which there has not been any effective surveys and thus classifying them as possibly extinct did not seem appropriate, including two Hawaiian yellow-faced bees (*Hylaeus facilis* and *Hylaeus hilaris*) (K. Magnacca, 2018, personal communication) and Fosberg's love grass (*Eragrostis fosbergii*) (*U.S. Fish and Wildlife Service, 2011*). If indeed extinct, all three were lost prior to protection under the ESA.

Including updated figures for number of listed species, time of protection and species extinctions, we estimate the ESA has prevented the extinction of roughly 291 species in its 45 year history. Based on the number of confirmed extinctions following listing, we further estimate that the ESA has to date prevented the extinction of more than 99% of species under its protection. To date, a total of 39 species have been delisted for recovery compared to four species that are extinct and 22 that are potentially extinct.

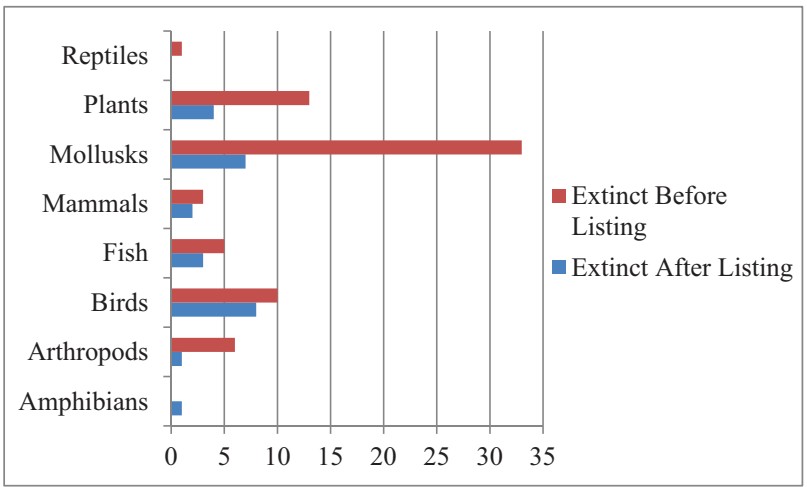

**Figure 1 Extinction and taxonomic group.** Extinct or possibly extinct listed species by taxonomic group.

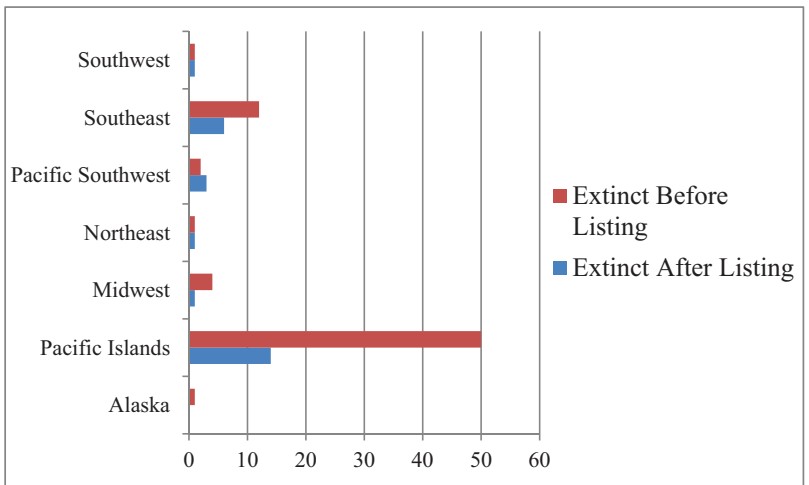

**Figure 2 Extinctions by region.** Extinct or possibly extinct listed species by U.S. Fish and Wildlife Service Region.

# DISCUSSION

The few number of listed species that have gone extinct following protection combined with an estimated 291 species for which extinction was prevented demonstrate the ESA has achieved one of its core purposes—halting the loss of species. We will not attempt to catalog them here, but numerous individual examples provide further support for this conclusion. Well known species like the California condor (*Gymnogyps californianus*), black-footed ferret (*Mustela nigripes*) and Hawaiian monk seal (*Neomonachus schauinslandi*), as well as lesser known species like the yellowfin madtom (*Noturus flavipinnis*), are but a few of the species that likely would have been lost were it not for the ESA.

The madtom is a case in point. Wrongly presumed extinct when described in 1969, individual madtom were found in the Powell River in Tennessee and Copper Creek in

Virginia and the species was protected under the ESA in 1977 (*U.S. Fish and Wildlife Service, 1977*). Following protection, federal and state officials worked with a non-governmental organization, Conservation Fisheries Inc., to discover additional populations and repatriate the species to rivers and streams in its historic range and there are now populations of the yellowfin madtom in three different watersheds (*U.S. Fish and Wildlife Service, 2012a*). The history of the ESA is replete with similar such stories.

The distribution of extinct or possibly extinct listed species largely tracks those regions with the highest rates of species endangerment, including Hawaii and the Northern Mariana Islands with 64 of the 97 extinctions or possible extinctions, and the Southeast with 18 of the extinctions or possible extinctions, mostly freshwater species. The fragility of Hawaii's endemic fauna to introduced species and habitat destruction and high degree of species imperilment is well recognized (*Duffy & Kraus, 2006*). Similarly, the extinction and endangerment of freshwater fauna in the southeast is well documented (*Benz & Collins, 1997*). To avoid further extinctions, these areas should be priorities for increased funding and effort.

Protection under the ESA came too late for the 71 species last seen prior to listing. It's possible that some of these species survived undetected following listing, but we find this unlikely for most if not all of the species. It is very difficult to document extinction, but all of the species were the subject of survey both before and after listing, which is described in the listing rules and subsequent status surveys. In addition, the 71 species were last seen an average of 24 years prior to listing, providing a long window for detection prior to listing. If some of these species did survive after listing it was likely at very low numbers, such that recovery would have been difficult at best.

That these 71 species were lost before protections were applied clearly highlights the need to move quickly to protect species. Indeed, *Suckling, Slack & Nowicki (2004)* identified 42 species that went extinct while under consideration for protection. Since that analysis was completed, the U.S. Fish and Wildlife Service has determined five additional species did not qualify for protection because they were extinct, including the Tacoma pocket gopher (*Thomomys mazama tacomensis*), Tatum Cave beetle (*Pseudanophthalmus parvus*), Stephan's riffle beetle (*Heterelmis* stephani), beaverpond marstonia (*Marstonia castor*) and Ozark pyrg (*Marstonia ozarkensis*), meaning there are now 47 species that have gone extinct waiting for protection (*U.S. Fish and Wildlife Service, 2012b*, *2016*, *2017*, *2018a*).

The U.S. Fish and Wildlife Service currently faces a backlog of more than 500 species that have been determined to potentially warrant protection, but which await a decision (*U.S. Fish and Wildlife Service, 2018b*). Under the ESA, decisions about protection for species are supposed to take 2 years, but on average it has taken the Fish and Wildlife Service 12 years (*Puckett, Kesler & Greenwald, 2016*). Such lengthy wait times are certain to result in loss of further species and run counter to the purpose of the statute. This problem can be addressed by streamlining the Service's process for listing species, which has become increasingly cumbersome, and by increasing funding for the listing program. For every species listed, the Service's process includes review by upward of 20 people, including numerous individuals who have no specific knowledge of the species and in a

number of cases are political appointees. We instead recommend that the Service adopt a process similar to scientific peer review, involving review by two to three qualified individuals.

The loss of 26 species after they were protected is indicative of conservation failure. This failure, however, in most cases cannot be wholly attributed to the ESA because most of these species were reduced to very low numbers by the time they were protected, making recovery difficult to impossible. Of the 19 species we could find an abundance estimate for at the time of listing, 13 had an estimated population fewer than 100 with eight having fewer than 10 individuals. Of the six other species, two Hawaiian birds, Oahu creeper (*Paroreomyza maculate*) and 'O'u (*Psittirostra psittacea*) had estimated populations in the hundreds, but this was based on sightings of single individuals. Given the lack of further sightings and the presence of disease carrying mosquitoes throughout their habitat, these estimates were likely optimistic. The other four species, the dusky seaside sparrow (*Ammodramus maritimus nigrescens*), Morro Bay kangaroo rat (*Dipodomys heermanni morroensis*), pamakani (*Tetramolopium capillare*) and Curtis' pearlymussel (*Epioblasma florentina curtisii*), had populations at the time of listing ranging from 100 to 3,000 individuals, but sufficient action was not taken to save them, making them true conservation failures.

At some level, all of the 97 ESA listed species that we identified as possibly extinct or extinct are conservation failures. For 42 of these species, the law itself was too late because they were last seen before the ESA was passed in 1973. But for others, there may have been time and we did not act quickly enough or dedicate sufficient resources to saving them. There are many examples of species both in the U.S. and internationally that have been successfully recovered even after dropping to very small numbers, but this can only occur with fast, effective action, resources and in many cases luck. The Mauritius kestrel (*Falco punctatus*), for example, was brought back from just two pairs (*Cade & Jones, 1993*) and the Hawaiian plant extinction prevention program, which focuses on saving plants with fewer than 50 individuals, has rediscovered many species believed extinct, brought 177 species into cultivation, constructed fences to protect species from non-native predators and reintroduced many species into the wild (*Wood, 2012*, http://www.pepphi.org/).

The failure to provide sufficient resources for conservation of listed species, however, continues to the present. As many as 27 species of Oahu tree snail (*achatinella spp.*) are extinct or possibly extinct, yet expenditures for the species that still survive are inadequate to support minimal survey and captive propagation efforts. Likewise, the Hawaiian plant extinction prevention program, which has been so effective in saving species on the brink of extinction, is facing a budget cut of roughly 70% in 2019 (http://www.pepphi.org/), which very likely could mean the extinction of dozens of plants that otherwise could be saved. Overall, *Greenwald et al. (2016)* estimate current recovery funding is roughly 3% of estimated recovery costs from federal recovery plans. We can save species from extinction, but it must be more of a priority for federal spending. Nevertheless, despite funding shortfalls and the tragedy of these species having gone extinct, the ESA has succeeded in preventing the extinction of the vast majority of listed species and in this regard is a success.

## Management implications

Of the 97 species we identified as extinct or potentially extinct, only 11 have been delisted for extinction. Another 11 have been recommended for delisting due to extinction. The San Marcos gambusia (*Gambusia georgei*) could also be delisted since there is very little hope it survives. For the other 74 possibly extinct species, we recommend retaining protections in the hope that some will be rediscovered and because there is little cost in retaining listing.

### Funding
The authors received no funding for this work.

### Competing Interests
All authors are employed by the Center for Biological Diversity which works to protect endangered species and their habitats.

### Author Contributions
- Noah Greenwald conceived and designed the experiments, performed the experiments, analyzed the data, prepared figures and/or tables, authored or reviewed drafts of the paper, approved the final draft.
- Kieran F. Suckling conceived and designed the experiments, performed the experiments, analyzed the data, authored or reviewed drafts of the paper, approved the final draft.
- Brett Hartl conceived and designed the experiments, performed the experiments, analyzed the data, authored or reviewed drafts of the paper, approved the final draft.
- Loyal A. Mehrhoff conceived and designed the experiments, performed the experiments, analyzed the data, authored or reviewed drafts of the paper, approved the final draft.

### Data Availability
The raw data are available in a Supplementary File and include a complete list of the species we identified as extinct or possibly extinct along with all supporting information.

### Supplemental Information
Supplemental information for this article can be found online at http://dx.doi.org/10.7717/peerj.6803#supplemental-information.

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
