# Peer review of "Extinction and the U.S. Endangered Species Act"

_PeerJ, doi:10.7717/peerj.6803_

## Round 0.1 · original submission · Major Revisions

While one of the reviewers is reasonably happy with this, the other (R2) raises substantial concerns about the analyses you've done. Please be sure to address these in your revision and in the cover letter that will accompany it. By all means contact me directly if you need to discuss any of the issue (s)he raises. I think these issues are clearly stated and I hope you can address them in ways that will satisfy the reviewer, to whom I shall send your revision.

Reviewer 1 ·

Basic reporting

Manuscript is written in a clear, and concise manner using professional language. Text is very direct ,easily understood and answers the questions posed by authors.. References cited are sufficient but more detaile discussions of P of survival of Hawaiian birds are available.

An easy read.

Experimental design

Suffient to the task and well defined.

Validity of the findings

Findings are defensible but I believe a bit conservative regarding status of Hawaiian birds e.g. Kamao Ou and Kauai Oo.Conclusions are well linked to original research question. They set up a more detailed follow up paper in which effort to locate "extinct species was quantified.

Additional comments

Greetings

Please consider the following comments
1) Lines 58-61 REcognizing that there will be wide variation in the answer discuss briefly what are expected time to recover after listing. For a quick assessment of the question you could use previously delistedy species as your data base.2) Page 2 lines 139 -142 provide reference

Reviewer 2 ·

Basic reporting

The basic reporting is good.

Experimental design

I have questions regarding the methodology, which are discussed in my review.

Validity of the findings

Due to my concerns about the methodology, I am not convinced of the findings. See details in my review.

Additional comments

This is a very important topic, and the authors are to be commended for tackling it in a quantitative fashion. However, contrary to the assertion on lines 72-73 that such a study has not been done previously, there was a paper addressing this same issue that reached some of the same conclusions (see: McMillan and Wilcove. 1994. Gone but not forgotten: why have species protected by the Endangered Species Act become extinct? Endangered Species UPDATE 11(11): 5-6). But it was published in an obscure journal, so it’s understandable why the writers have overlooked it. Moreover, this manuscript by Greenwald et al. is more comprehensive and also attempts to calculate the number of species whose extinctions have been prevented by the ESA.

With respect to the first part of the manuscript – determining how many and which species have gone extinct before ESA listing and after ESA listing – I have the following questions:

1. The authors state that 71 of 97 extinct (EX) or possible extinct (PE) species were last observed before they were listed and thus “are not relevant” to determining the Act’s success in preventing extinctions. These 71 species were last seen anywhere from 1-80 years prior to listing. It strikes me that unless there were fairly intensive searches of the habitat immediately prior to or immediately after listing, it is not reasonable to exclude these species from an assessment of the Act’s success. That a species was last seen 1, 2, or 10 years prior to listing is, in itself, uninformative, unless we know that serious efforts were made to find it right before or after listing. The authors may well have factored this into their determinations but it is unclear from the text and SI that they did so.
2. Of the 26 species that disappeared after listing, the authors note that 8 had total populations <10 individuals at time of listing that their survival was always questionable. This could be fleshed out more: what was the distribution of population sizes at time of listing for these 26 species?

With respect to the second part of the manuscript – calculating how many species extinctions have been averted by ESA listing – I have several concerns regarding the methodology:

1. An underlying assumption of the methodology is that the ESA designations of Endangered and Threatened can be equated to the IUCN Red List designation of endangered, with the associated probability of persistence. This cannot be assumed; it must be tested. The Red List criteria are based on quantitative criteria related to population size and number, rate of decline, and or of habitat loss, which (allegedly) are they linked to quantitative assessments of probability of extinction as assessed via PVAs. However, the statutory definitions in the ESA of “Endangered” and “Threatened” are notably vague; nor, to the best of my knowledge, has FWS set about establishing quantitative guidelines as what puts a species in one status versus the other. In the absence of such guidance, the authors need to look at the population sizes or rates of habitat loss of T&E species and see if those values more or less align with the IUCN criteria.
2. The equation they use for determining the null model – how many species would have disappeared in the absence of ESA protection – is as follows:

Expected extinctions = (Spp. X 100 year Extinction Risk X Portion of a century with protection)

Are they using 45 years as the numerator in determining the proportion of 67 years that a species has enjoyed ESA protection? If so, this puzzles me because species only get that protection once they are listed, and they have been added to the list at varying times. Thus, should not the null model use this equation and make a separate calculation for each species based on how long it has enjoyed ESA protection and then sum the results to derive the expected figure for how many species should have perished in the absence of ESA protection? Perhaps this is what they did, although I cannot find confirmation of this in the material sent to me. (And, of course, this does not negate my belief that the foundation of the calculation may be invalid for the reason outlined in the previous point).

Finally, the authors allude (lines 186-7) to steps FWS can take to streamline the listing process. If space permits, it would be great to provide at least a summary of these proposed changes.

---

## Round 0.2 · accepted · Accept

I think you've done an excellent job in responding to my reviewers' concerns and do not see the need to return it to them for further comments. Thanks so much for sending this to PeerJ, where we now have a good record for publishing key papers on conservation.

#